

# Huperzine A attenuates nonalcoholic fatty liver disease by regulating hepatocyte senescence and apoptosis: an in vitro study

Xiao-na Hu[1,2,3,*], Jiao-feng Wang[2,3,*], Yi-qin Huang[1,2,*], Zheng Wang[2,3], Fang-yuan Dong[2,3], Hai-fen Ma[2] and Zhi-jun Bao[1,2,3]

[1] Department of Gastroenterology, Huadong Hospital Affiliated to Fudan University, Shanghai, China
[2] Shanghai Key Laboratory of Clinical Geriatric Medicine, Shanghai, China
[3] Department of Geriatrics, Huadong Hospital Affiliated to Fudan University, Shanghai, China
* These authors contributed equally to this work.

## ABSTRACT

**Objective:** This study was undertaken to detect if free fatty acids (FFA) induce hepatocyte senescence in L-02 cells and if huperzine A has an anti-aging effect in fatty liver cells.

**Methods:** L-02 cells were treated with a FFA mixture (oleate/palmitate, at 3:0, 2:1, 1:1, 1:2 and 0:3 ratios) at different concentrations. Cell viability and fat accumulation rate were assessed by a Cell Counting Kit 8 and Nile Red staining, respectively. The mixture with the highest cell viability and fat accumulation rate was selected to continue with the following experiment. The L-02 cells were divided into five groups, including the control group, FFA group, FFA + 0.1 μmol/L huperzine A (LH) group, FFA + 1.0 μmol/L huperzine A (MH) group and FFA + 10 μmol/L huperzine A (HH) group, and were cultured for 24 h. The expression of senescence-associated β-galactosidase (SA-β-gal) was detected by an SA-β-gal staining kit. The expression levels of aging genes were measured by qRT-PCR. The expression levels of apoptosis proteins were detected by a Western blot. ELISA kits were used to detect inflammatory factors and oxidative stress products. The expression of nuclear factor (NF-κB) and IκBα were detected by immunofluorescence.

**Results:** The FFA mixture (oleate/palmitate, at a 2:1 ratio) of 0.5 mmol/L had the highest cell viability and fat accumulation rate, which was preferable for establishing an in vitro fatty liver model. The expression of inflammatory factors (TNF-α and IL-6) and oxidants Malonaldehyde (MDA), 4-hydroxynonenal (HNE) and reactive oxygen species (ROS) also increased in the L-02 fatty liver cells. The expression levels of aging markers and aging genes, such as SA-β-gal, p16, p21, p53 and pRb, increased more in the L-02 fatty liver cells than in the L-02 cells. The total levels of the apoptosis-associated proteins Bcl2, Bax, Bax/Bcl-2, CyCt and cleaved caspase 9 were also upregulated in the L-02 fatty liver cells. All of the above genes and proteins were downregulated in the huperzine A and FFA co-treatment group. In the L-02 fatty liver cells, the expression of IκBα decreased, while the expression of NF-κB increased. After the huperzine A and FFA co-treatment, the expression of IκBα increased, while the expression of NF-κB decreased.

Corresponding author
Zhi-jun Bao,
zhijunbao@fudan.edu.cn

**Conclusion:** Fatty liver cells showed an obvious senescence and apoptosis phenomenon. Huperzine A suppressed hepatocyte senescence, and it might exert its anti-aging effect via the NF-κB pathway.

## INTRODUCTION

Currently, nonalcoholic fatty liver disease (NAFLD) is the most prevalent liver disease, and it is a major public health hazard. The prevalence of NAFLD in western countries is variable, ranging from 24–46% (*Browning et al., 2004*; *Williams et al., 2011*; *Caballeria et al., 2010*). The estimated prevalence of NAFLD in Asia is widely variable, ranging from 7.9–54.0% (*Seto & Yuen, 2016*). NAFLD comprises a histological spectrum, ranging from nonalcoholic fatty liver, nonalcoholic steatohepatitis (NASH), NASH-related cirrhosis and hepatocellular carcinoma (HCC) (*Calzadilla & Adams, 2016*). Studies report that almost 10–20% of individuals with NAFLD have NASH, and 5–25% of individuals with NASH have the potential to progress to cirrhosis. NASH-related cirrhosis, HCC and liver failure have become the leading cause of morbidity and mortality in patients (*Hashimoto & Tokushige, 2011*; *Calzadilla & Adams, 2016*; *Hu et al., 2012*).

The two-hit hypothesis is a wildly accepted mechanism of NAFLD. Fat deposition in the cytoplasm of the hepatocytes is claimed to be the first hit. Oxidative stress, lipid peroxidation and inflammation serve as the second hit, which contributes to the progression of steatosis to steatohepatitis. Increasingly more evidence indicates that age is an independent predictor of NAFLD (*Bertolotti et al., 2014*). The prevalence of NAFLD increases with increasing age (*Hu et al., 2012*; *Kim, Kisseleva & Brenner, 2015*). Older age can increase the likelihood of NASH and fibrosis. Older age also increases the risk of death of those with NAFLD (*Wang et al., 2013*). Animals models of NAFLD have also revealed features of accelerated aging. Hepatocyte senescence may serve as a third hit to the progression of NAFLD. However, how hepatocyte senescence occurs is unclear.

Recent studies have reported that there is a close association between oxidative stress and cellular senescence (*Zhang et al., 2017*; *Venkatachalam, Surana & Clement, 2017*). To study the role of oxidative stress in hepatocyte senescence, we selected an antioxidative agent, huperzine A, to treat an in vitro NAFLD model and tested the status of hepatocyte senescence.

Huperzine A, a selective AChE inhibitor, is widely used to treat Alzheimer's disease (*Yang et al., 2013*). However, currently, many studies have revealed that it has antioxidative and anti-inflammatory effects. Huperzine A inhibits the overexpression of pro-inflammatory factors in a rat model of transient focal cerebral ischemia (*Wang et al., 2008*). It also upregulates the cholinergic anti-inflammatory pathway (*Brenner et al., 2008*), suppresses immunological cell proliferation and inhibits age-related disorders (*Pollak et al., 2005*; *Brenner et al., 2008*; *Tabet, 2006*; *Ruan et al., 2013*). Our previous study also documented that huperzine A suppressed D-gal-induced neurovascular damage

and blood–brain barrier dysfunction by preventing the nuclear factor-κB (NF-κB) pathway (*Ruan et al., 2013*). Therefore, we conducted an in vitro study to find the relationship among inflammation, oxidative stress and hepatocyte senescence and to determine if huperzine A exerts its antioxidative and anti-aging effects via the NF-κB pathway in NAFLD.

## MATERIALS AND METHODS

### Cell culture

The L-02 cells were purchased from The Academy of Sciences of China. The L-02 cells were cultured in RPMI 1640 medium (Gibco, Carlsbad, CA, USA) supplemented with 10% fetal bovine serum (Gibco, CA, USA), 100 U/mL penicillin and 100 U/mL streptomycin (Gibco, Carlsbad, CA, USA) for 24 h. All the cell lines were maintained at 37 °C in a humidified incubator with an atmosphere of 5% $CO_2$ (*Qin, Yin & Huang, 2016*).

### Preparation of FFA culture medium

Sodium palmitate and oleate (Sigma, New York, NY, USA) were added to the $H_2O_2$ and NaOH solutions, respectively, and then, they were added to a 10% solution of free fatty acids (FFA) bovine serum albumin (BSA) (Sangon Biotech, Shanghai, China), which was dissolved overnight at 55 °C with gentle shaking; then, the mixture was cooled to room temperature, filtered and stored at 4 °C in a refrigerator (*Qin, Yin & Huang, 2016*).

### Establishment of L-02 fatty liver cell model

Stock solutions of 50 mmol/L sodium oleate and 50 mmol/L sodium palmitate were mixed at five different ratios (oleate/palmitate, at 3:0, 2:1,1:1, 1:2 and 0:3 ratios) and were then conveniently diluted in culture medium containing 1% BSA to obtain the desired final concentrations (0.25, 0.5, 0.75, 1.0 and 1.5 mmol/L) (*Shi et al., 2014*; *Gomez-Lechon et al., 2007*). To induce fat-overloading in cells, the L-02 cells were serum-starved overnight and were then incubated with different permutations of the concentrations and ratios of the FFA mixture for 24 h. The cells treated with culture medium containing 1% BSA were used as the control group. Each group was established in three batches and was recultured six times (*Gomez-Lechon et al., 2007*).

### Cell viability assay

The cells were seeded in 96-well plates at a concentration of $1.0 \times 10^4$ cells/well. Then, the cells were maintained at 37 °C for 24 h. Cell viability was determined with Cell Counting Kit 8 (CCK8) (Dojindo, Xiongben, Japan) following the manufacturer's instructions. After treatment with FFA mixture for the times and concentrations indicated, cell viability was assessed by incubation with 10 μl of tetrazolium substrate for 1 h at 37 °C, followed by measurement of absorbance at 450 nm (*Chugh et al., 2012*).

### Nile Red staining

The L-02 cells were seeded at a density of $5.0 \times 10^4$ cells/well and were exposed to the mixture of sodium oleate and sodium palmitate as described above. The cells were washed

twice with phosphate buffer saline (PBS) and were then stained with the Nile Red solution at a concentration of 1 mg/L in PBS for 15 min at 37 °C. Then, the cell monolayers were washed with PBS. The images were recorded at 543-nm excitation and 598 nm emission wavelengths using fluorescence microscopy (Olympus, Tokyo, Japan) to analyze the accumulation rate of the lipids in the L-02 cells (*McMillian et al., 2001*).

## Experimental design of the effects of huperzine A on L-02 fatty cells

A mixture of sodium oleate and sodium palmitate, mixed at a concentration of 0.5 mmol/L with a ratio of 2:1 (oleate/palmitate), was selected as the optimum FFA mixture. The L-02 cells were seeded in six-well plates at a density of $2.0 \times 10^5$ cells/well. Then, the L-02 cells were divided into five groups, including the control group, FFA group, FFA+0.1 μmol/L huperzine A (LH) group, FFA + 1.0 μmol/L huperzine A (MH) group and FFA + 10 μmol/L huperzine A (HH) group. In the huperzine A treatment group, the L-02 cells were pretreated with RPMI 1640 containing different concentrations of huperzine A for 12 h, after which the FFA mixture was added to the culture system, with exception of the control group; the groups were then incubated for a further 24 h.

## Flow cytometric analysis of cell apoptosis

Apoptotic cells were assessed using the Annexin V-fluorescein isothiocyanate (FITC) apoptosis detection kit I (BD Bioscience, Franklin, NJ, USA) following the manufacturer's instructions. L-02 cells in each group (mentioned above) were digested with trypsinization after treated for 24 h. Detached cells were collected by centrifugation at 1,500 rpm for 6 min. Then the harvested cells were washed by PBS for twice, resuspended in the binding buffer, and incubated with Annexin V-FITC and propidium iodide staining solution. Samples of 10,000 stained cells were analyzed using a flow cytometer (BD Bioscience, Franklin, NJ, USA) (*Chu et al., 2011*).

## Measurement of hepatocyte senescence markers

A senescence-associated β-galactosidase staining (SA-β-gal) kit (Beyotime Institute of Biotechnology, Suzhou, China) was used for SA-β-gal staining. The cells in each group were washed three times using PBS and were then fixed in the SA-β-gal fixing solution for 15 min after removing the cell-culture medium. Then, the cells were again washed three times in PBS and were incubated overnight at 37.0 °C with the SA-β-gal working solution. The cells were stained with DAPI for 1 h and were then dehydrated and stored at 4 °C. The images were captured at 200× magnification using a regular microscope and a fluorescence microscope (Olympus, Tokyo, Japan). The data are expressed as the mean ± standard deviation (SD) of at least three independent experiments for each experiment. Each sample was repeated three times.

## Analysis of proliferation

For the Edu (RiboBio, Guangzhou, China) immunofluorescence, the L-02 cells were seeded in 24-well plates with cover slips at a concentration of $2.0 \times 10^4$ cells/well and were maintained at 37 °C for 24 h. A total of 10 μm Edu was added to the 24-well plate, and it was then co-incubated with the cells for 4 h. Then, the cells were fixed with cold

4% paraformaldehyde in 0.1 M PBS for 30 min and were rinsed with PBS three times. Next, the cells were incubated with 0.2% Triton X-100 for 10 min and were then rinsed with PBS three times. The Edu antibody was diluted with cell-culture medium at a ratio of 1:1,000, and 300 $\mu$L of the solution was added to each well. The cells were incubated at 4 °C overnight and rinsed with PBS three times, and the nuclei were stained using 0.5 $\mu$g/mL DAPI. The images were captured using a regular microscope and a fluorescence microscope (Olympus, Tokyo, Japan). Each sample was repeated three times.

## ELISA analysis of oxidation, antioxidation and inflammation

The L-02 fatty cells were treated by huperzine A for 24 h, and then, the cell culture supernatants were collected. TNF-$\alpha$ and IL-6 ELISA kits (Proteintech, Chicago, IL, USA) were used to measure the expression of the inflammatory factors TNF-$\alpha$ and IL-6. Malonaldehyde (MDA), 4-hydroxynonenal (HNE) and reactive oxygen species (ROS) kits (Beyotime Institute of Biotechnology, Suzhou, China) were used to measure the expression of MDA, HNE and ROS to analyze oxidative stress. The content of the inflammatory factors and oxidative stress products above were measured according to the kit instructions. Each sample was measured in triplicate.

## Western blot analysis

The cells were lysed with RIPA lysis buffer (Beyotime, Suzhou, China) with protease inhibitor cocktails (Sigma, New York, NY, USA) on ice for 5 min. Then, the lysates were centrifuged at 12,000 rpm for 10 min at 4 °C, and the supernatant was collected in a new fresh EP tube. Protein was extracted with chilled lysis buffer containing 1 mmol/L PASF. The protein concentration was detected using a BCA protein assay kit (Beyotime, Suzhou, China). The protein samples were separated on SDS–PAGE gels and were then transferred to PVDF membranes. The membranes were first probed with primary antibodies overnight at 4 °C and were then incubated with an HRP-conjugated goat anti-rabbit/mouse secondary antibody at room temperature for 1 h; then, the blots were detected using the ECL system. The primary antibodies were $\beta$-actin (diluted 1:1,000), 8-oxoG (diluted 1:1,000), Bcl-2 (diluted 1:2,000), Bax (diluted 1:1,000), cytochrome C (CytC, diluted 1:1,000), cleaved caspase 9 (diluted 1:1,000) and procaspase 9 (diluted 1:1,000). The secondary antibodies were goat anti-rabbit and goat anti-mouse that were HRP-conjugated (diluted 1:10,000). The GAPDH, 8-oxoG, CytC, cleaved caspase 9 and procaspase 9 antibodies were purchased from Proteintech (Chicago, IL, USA), and Bcl-2 and Bax were from Cell Signaling Technology (Boston, MA, USA). The secondary antibodies were from Jackson (Pennsylvania, USA).

## Real-time Quantitative Reverse Transcription PCR analysis

The cells were rinsed with PBS three times after removing the culture medium. Total RNA was extracted from the experimental cells with Trizol (Invitrogen, Carlsbad, CA, USA) for cDNA synthesis. A total of 2 $\mu$g of RNA was used to generate cDNA with the SMART® MMLV Reverse Transcriptase kit (Takara, Mountain View, CA, USA). For the RT-PCR, the cDNA and primers were prepared with a SYBR Green qPCR

SuperMix Kit (Takara, Tokyo, Japan), according to the instruction manual. GAPDH was used as a reference gene. The following primers were used: p16, forward 5′-CCTGGAGGCGGCGAGAACAT-3′ and reverse 5′-CGCGGGATGTGAACCACGAAA-3′; p53, forward 5′-CCTCCTCAGCATCTTATCCG-3′ and reverse 5′-CAACCTCAGGCGGC TCATAG -3′; p21, forward 5′-ACCGAGACACCACTGGAGGG-3′ and reverse 5′- CC TGCCTCCTCCCAACTCATC-3′; pRb, forward 5′-CCAGGCGAGGTCAGAACAGG-3′ and reverse 5′-CCTCTGGAAGTCCATTAGATGTTAC-3′; and GAPDH, forward 5′- AATCCCATC ACCATCTTCCAG-3′ and reverse 5′-ATCAGCAGAGGGGGGCAGAGA-3′. The primers above were designed using Primer Express software 5.0 (Applied Biosystems, Foster City, CA, USA). They were synthesized by Suzhou Genewiz Biology Company (Suzhou, China). One times cDNA template, two times cDNA template and 0.5 times cDNA template were used to detect dose-effect relationship of p16, p53, p21, pRb and GAPDH respectively on ABI 7500 fluorescence quantitative PCR apparatus. The results showed that the quantitative relationship between the fluorescence of PCR products and the cDNA template was $R^2 > 0.99$, and their line slope were comparable, which proved that these primers were appropriate and the experimental results based on this method were accurate and credible. Thus we calculated relative fold changes in gene expression normalized to GAPDH by the $\Delta\Delta$CT method using the equation $2^{-\Delta\Delta CT}$. The amplification conditions were as follows: 95 °C for 5 min for initial denaturation, followed by denaturation at 95 °C for 10 s, and a combined annealing and extension at 60 °C for 30 s through 40 amplification cycles. Each sample was measured in triplicate.

## Immunofluorescence

The L-02 cells were seeded in 24-well plates with cover slips at a concentration of $2.0 \times 10^4$ cells/well and were maintained at 37 °C for 24 h; then, the cells were rinsed with PBS three times, 3 min each time. Then, the cells were fixed with 4% paraformaldehyde for 15 min and rinsed with PBS three times, 3 min each time. Next, the cells were incubated with 0.5% Triton X-100 for 20 min at room temperature and were then rinsed with PBS three times, 3 min each time. The PBS on the cover slips was dried with wipes, and normal goat serum was added on the cover slips. They were incubated for 30 min at room temperature. The goat serum on cover slips was dried with wipes, and then, the cover slips were first probed with primary antibodies overnight at 4 °C in a wet box. Then, the cover slips were rinsed with Phosphate Buffered Solution with Tween-20 (PBST) three times. The cover slips were dried with wipes and were incubated with secondary antibodies for 1 h at room temperature. Then, the cover slips were rinsed with PBST three times (3 min each time), and the nuclei were stained with DAPI. Then, the cover slips were rinsed with PBST 4 times (5 min each time), and the cover slips were dried with wipes. The cover slips were treated by antifade mounting medium. The images were captured using a fluorescence microscope (Olympus, Tokyo, Japan). The primary antibodies were NF-κB p65 (diluted 1:100) (Proteintech, Chicago, IL, USA), IκBα (diluted 1:100) (Proteintech, Chicago, IL, USA), and NF-κB p-p65 (diluted 1:100) (Cell Signaling Technology, Boston, MA, USA), and the secondary antibodies were goat anti-mouse with HRP-conjugated (diluted 1:10,000).
## Statistical analysis

Data are presented as the means ± SD for at least three independent experiments for every group. Differences among groups were all compared using a one-way ANOVA, Fisher's LSD test and fixed effects model. All the data were analyzed using GraphPad Prism (GraphPad Software, San Diego, CA, USA) and SPSS 18.0 (SPSS Inc., Chicago, IL, USA). $P < 0.05$ was considered a statistically significant difference.

# RESULTS

## Effects of FFA on the viability of the L-02 cells

To determine the cytotoxic effects of FFA, CCK8 was selected to test the viability of the L-02 cells incubated with the FFA mixture at various ratios (oleate/palmitate at 3:0, 2:1,1:1, 1:2 and 0:3 ratio) and concentrations (0.25, 0.5, 0.75, 1.0 and 1.5 mmol/L) after 24 h of treatment. The results showed that the cell viabilities were different with different ratios of the FFA mixture. The FFA mixture (oleate/palmitate) produced a significant toxicity at those ratios with a higher palmitic acid content (1:1, 1:2 and 0:3 ratios) ($P < 0.05$). The viability of the L-02 cells incubated with FFA mixed at a ratio of 2:1 (oleate/palmitate) was higher than that for the other ratios ($P < 0.05$). The viability of the L-02 cells incubated with 0.25, 0.5, 0.75 and 1.0 mmol/L of the FFA mixture (oleate/palmitate) were all higher than those incubated with 1.5 mmol/L of the FFA mixture ($P < 0.05$) (Fig. 1A).

## Fat overloading of hepatic cells with different concentrations of FFA

Observation from both the white light microscopic image and fluorescent microscopic image showed that there was significant fat accumulation in the L-02 cells incubated with the FFA mixture. The mixture with a concentration of 0.5 mmol/L of FFA (oleate/palmitate, at a 2:1 ratio) had a higher lipid accumulation rate than did the other concentrations of FFA ($P < 0.05$) (Figs. 1B–1C). These results demonstrated that the 0.5 mmol/L of FFA mixture, with oleate and palmitate mixed a ratio of 2:1 (oleate/palmitate) had a higher cell viability and higher fat accumulation, which could be used to establish the L-02 fatty liver model.

## Fatty liver cells show a senescence phenomenon and huperzine A weakens cell senescence

The optimum FFA mixture, at a concentration of 0.5 mmol/L and a ratio of 2:1 (oleate/palmitate), was selected to establish the L-02 fatty liver model in the following experiments. To determine the cell senescence status, immunofluorescence labeling for SA-β-gal was selected to analyze the number of aging cells. The results showed that the expression of the SA-β-gal-positive cells was nearly threefold in the FFA group compared with in the control group under fluorescence microscopy ($P < 0.01$). The expression levels of the SA-β-gal-positive cells in the FFA + LH, FFA + MH and FFA + HH groups were all lower than those in the FFA group ($P < 0.01$). The expression levels of the SA-β-gal-positive cells in FFA + HA group were the lowest ($P < 0.01$) (Figs. 2A–2F).

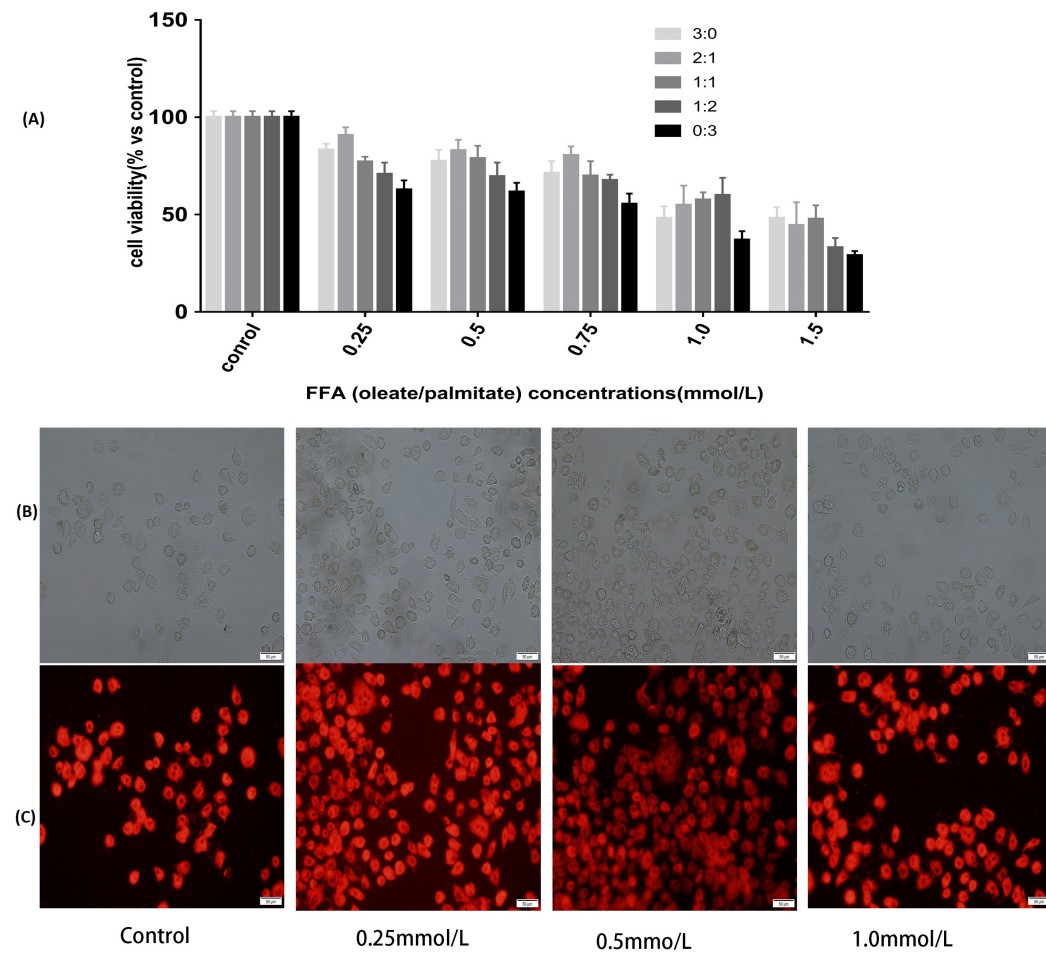

**Figure 1 Effects of FFA on cell viability and lipids accumulation in L-02 cells.** (A) L-02 cells were treated with FFA mixture at various ratios (oleate/palmitate at 3:0, 2:1, 1:1, 1:2 and 0:3 ratio) and concentrations (0.25, 0.5, 0.75, 1.0 and 1.5 mmol/L) for 24 h. Cell viability was determined by a Cell Counting Kit 8 (CCK8) kit. The viability of the L-02 cells incubated with FFA mixed at a ratio of 2:1 (oleate/palmitate) was higher than that for the other ratios in FFA lower than 1.0 mmol/L groups ($P < 0.05$). The viability of the L-02 cells incubated with 0.25, 0.5, 0.75 and 1.0 mmol/L of the FFA mixture (oleate/palmitate) were all higher than those incubated with 1.5 mmol/L of the FFA mixture ($P < 0.05$). (B) The white light microscopic image of lipid dropets. (C) The fluorescent microscopic image of lipid dropts (dyed red in Nile Red staining). The FFA mixture of 0.5 mmol/L with sodium oleate and sodium palmitate mixed a ratio of 2:1 had higher lipid accumulation rate ($P < 0.05$).

These results suggested that L-02 fatty liver cells showed a senescence phenomenon. Huperzine A suppressed the cell senescence induced by FFA in a dose-dependent manner.

## Huperzine A improves the cell proliferation activity and viability and inhibits apoptosis in L-02 fatty liver cells

To confirm whether the cell proliferation activity decreased, an immunofluorescence assay of Edu was selected to analyze cell proliferation. The results showed that the expression of Edu in the FFA group was lower than in the control group ($P < 0.01$). The expression levels were all higher in FFA + LH, FFA + MH and FFA + HH groups than

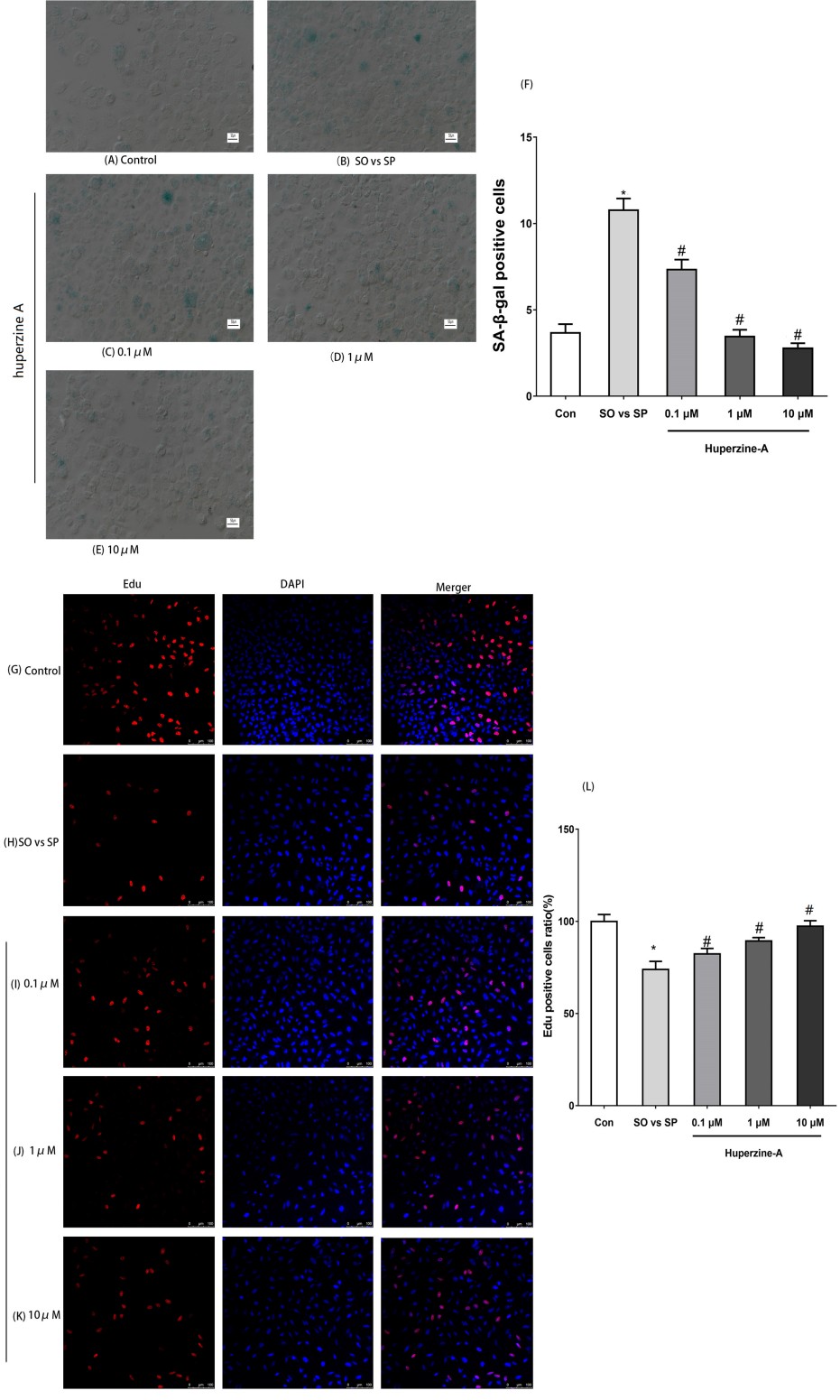

(F)

(L)

**Figure 2 The expression of SA-β-gal and Edu by immunofluorescence labeling.** (A–F) L-02 cells were treated with FFA, FFA + LH, FFA + MH and FFA + HH for 24 h. The expression of SA-β-gal determined by immunofluorescence labeling. Quantitative analysis of SA-β-gal-positive cells in the five groups by the ratio of SA-β-gal-positive cell number to the total and normalized to the controls. The expression of the SA-β-gal-positive cells was nearly threefold in the FFA group compared with in the control group ($P < 0.01$), but reversed in the FFA + LH, FFA + MH and FFA + HH groups ($P < 0.01$). (G–L) The expression of Edu was detected by immunofluorescence labeling. Quantitative analysis of Edu-positive cells in the five groups by the ratio of Edu-positive cell number to the total and normalized to the controls. The expression of Edu in the FFA group was lower than in the control group ($P < 0.01$). The expression levels were all higher in FFA + LH, FFA + MH and FFA + HH groups than in the FFA group ($P < 0.01$), but with no significant difference among FFA + LH, FFA + MH and FFA + HH groups ($P > 0.05$). Data presented as the mean ± SE ($n = 3$ independent replicates). *$P < 0.05$ vs. control; #$P < 0.05$ vs. FFA models; FFA mixture: 0.5 mmol/L, sodium oleate (SO)/sodium palmitate (SP) mixed at a 2:1 ratio; LH: FFA + 0.1 μmol/L huperzine A; MH:FFA + 1.0 μmol/L huperzine A; HH:FFA + 10 μmol/L huperzine A.

in the FFA group ($P < 0.01$), but with no significant difference among FFA + LH, FFA + MH and FFA + HH groups ($P > 0.05$) (Figs. 2G–2L).

Flow cytometry was used to detect cell apoptosis. The results showed that the percentage of apoptotic cells was increased from 3.127% ± 0.308% in control cells to 8.800% ± 0.089% in FFA-treated cells ($P < 0.01$). The percentage of apoptotic cells were 5.690% ± 0.405%, 5.263% ± 0.276% and 3.640% ± 0.907% in LH, MH and HH groups, respectively, which were all lower than in the FFA group ($P < 0.05$) (Figs. 3A–3B). These results revealed that FFA decreased the cell proliferation activity and viability, but huperzine A reversed the role of FFA in reducing the cell proliferative activity and viability.

To determine if FFA induced cell apoptosis, we tested the apoptosis proteins 8-oxoG, Bcl2, Bax, CytC and cleaved caspase 9 using a Western blot. The results showed that the total protein levels of 8-oxoG, Bax, Bax/Bcl2, CytC and cleaved caspase 9 were upregulated in the FFA group compared with the levels in the control group ($P < 0.01$), while the protein levels of Bcl2 were downregulated in the FFA group compared with the levels in the control group ($P < 0.01$). The expression levels of 8-oxoG, Bax, Bax/Bcl2, CytC and cleaved caspase 9 decreased in the FFA + LH, FFA + MH and FFA + HH groups compared with those of the FFA group, but with no significant difference among FFA + LH, FFA + MH and FFA + HH groups ($P > 0.05$) (Figs. 3C–3I). These results demonstrated that FFA induced cell apoptosis, while huperzine A attenuated the cell apoptosis induced by FFA.

## Huperzine A deregulated senescence genes, inflamm-aging factors and oxidative stress

To verify the effects of huperzine A on cell senescence, qRT-qPCR was used to analyze the mRNA levels of the aging genes p16, p21, p53 and pRb. The results showed that the expression of p16, p21, p53 and pRb in the FFA group was higher than in the control group ($P < 0.01$) (Figs. 4A–4D). All of the mRNA levels of above aging genes in the FFA + LH, FFA + MH and FFA + HH groups were decreased compared with those in the FFA group ($P < 0.01$) (Figs. 4A–4D). To analyze the expression of the inflamm-aging factors,

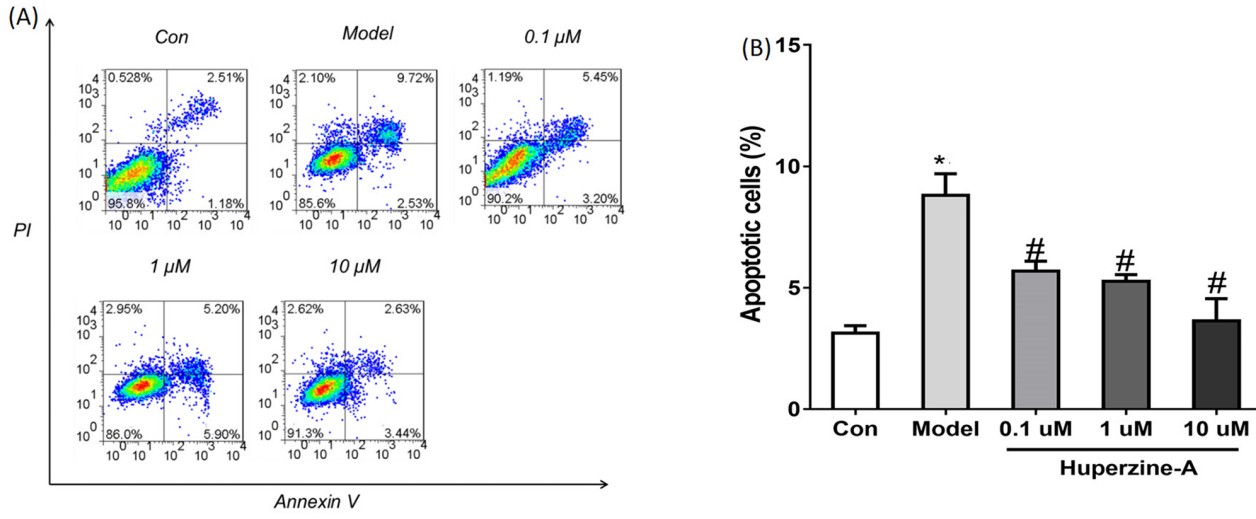

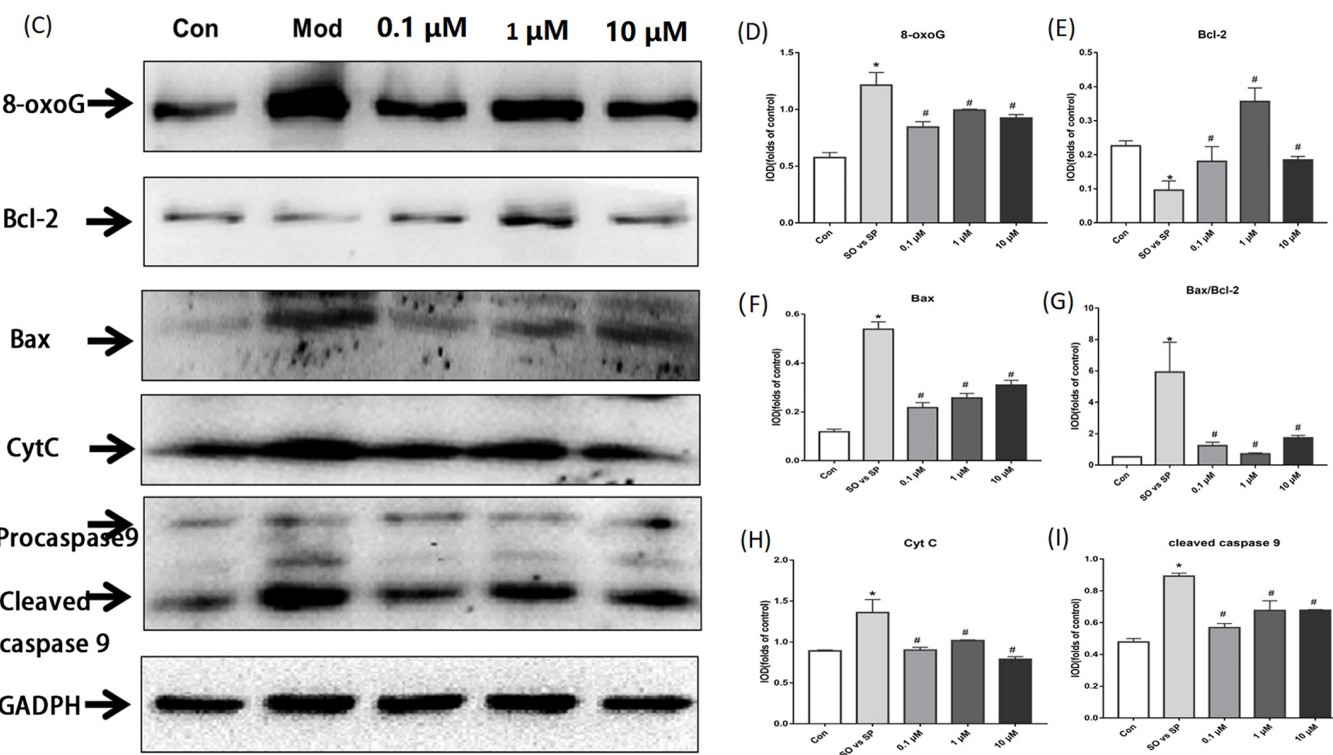

**Figure 3 Cell apoptosis and the expression of apoptosis proteins determined by flow cytometry and Western Blot.** (A and B) L-02 cells were treated with FFA, FFA + LH, FFA + MH and FFA + HH for 24 h and stained with Annexin V-fluorescein isothiocyanate (FITC) and propidium iodide. Apoptotic and necrotic cells were monitored by flow cytometry. The percentage of apoptotic cells was higher in FFA group than in the controls ($P < 0.01$). The percentage of apoptotic cells in LH, MH and HH groups were lower than in the FFA group ($P < 0.05$). (C–I) The expression of apoptosis proteins was determined by Western Blot. The total protein levels of 8-oxoG, Bax, Bax/Bcl2, CytC and cleaved caspase 9 were upregulated in the FFA group compared with the levels in the control group ($P < 0.01$). They were decreased in the FFA + LH, FFA + MH and FFA + HH groups compared with those of the FFA group ($P < 0.01$), but with no significant difference among FFA + LH, FFA + MH and FFA + HH groups ($P > 0.05$). Data presented as the mean ± SE ($n = 3$ independent replicates). *$P < 0.01$ vs. control; #$P < 0.01$ vs. FFA models; FFA mixture: 0.5 mmol/L, sodium oleate (SO)/sodium palmitate (SP) mixed at a 2:1 ratio; LH: FFA + 0.1 µmol/L huperzine A; MH: FFA + 1.0 µmol/L huperzine A; HH: FFA + 10 µmol/L huperzine A.

an ELISA was selected to test TNF-α and IL-6. The results showed that the protein levels of TNF-α and IL-6 in the FFA group were higher than in the control group ($P < 0.01$) (Figs. 4E–4F). The expression levels of these factors were decreased in the FFA + LH, FFA + MH and FFA + HH groups compared with those in the FFA group ($P < 0.01$), but with no significant difference among FFA + LH, FFA + MH and FFA + HH groups ($P > 0.05$) (Figs. 4E–4F). To further analyze the status of oxidative stress, an ELISA was used to test the expression of oxidation factors, such as MDA, 4-HNE and ROS. The results showed that the levels of MDA, 4-HNE and ROS were increased in the FFA group compared with those in the control group ($P < 0.01$), and they decreased in the FFA + LH, FFA + MH and FFA + HH groups compared with those in the FFA group ($P < 0.01$) (Figs. 4G–4I). In addition, they decreased with increasing huperzine A concentrations ($P < 0.05$) (Figs. 4G–4I). These results revealed that FFA upregulated the expression of senescence genes, inflamm-aging factors and oxidative stress, while huperzine A downregulated these factors in a concentration-dependent manner.

**Huperzine A might suppress L-02 fatty liver cell senescence via the NF-κB pathway**

To further confirm whether the NF-κB pathway plays an important role in cell senescence, we measured the levels of NF-κB-p-p65 and IκBα by an immunofluorescence assay. The mean optical density (MOD) value was used to analyze the quantitative of NF-κB-p-p65 and IκBα. The results showed that the MOD value of IκBα was decreased in the FFA group compared with that of the control group ($P < 0.01$) (Figs. 5A–5B). The MOD value of IκBα increased in the FFA + LH, FFA + MH and FFA + HH groups compared with that in the FFA group ($P < 0.01$) (Figs. 5A–5B). The MOD value of NF-κB-p-p65 in the FFA group was higher than in the control group, but was lower in the FFA + LH, FFA + MH and FFA + HH groups than in the FFA group ($P < 0.01$) (Figs. 5C–5D). There was no difference of MOD values for the two factors in different huperzine A concentration ($P > 0.05$). These results suggested that the NF-κB-p-p65 pathway was activated after FFA treatment. Huperzine A might suppress cell senescence induced by FFA via the NF-κB pathway.

## DISCUSSION

The pathogenesis of NAFLD is widely interpreted by the "two-hit" hypothesis. Lipid metabolism disorder, such as an increased concentration of FFA and a disturbed fatty acid oxidation process, serve as the first hit in the liver. The treatment of HepG2 and L-02 cell lines with monounsaturated and saturated fatty acids is commonly used to make an in vitro NAFLD model and to study the mechanism of hepatic steatosis (*Gomez-Lechon et al., 2007*). In our study, we treated L-02 cells with sodium oleate and palmitate and found that oleate and palmitate mixed at 0.5 mmol/L, with a ratio of 2:1, could be used to establish a suitable in vitro NAFLD model.

The second hit is correlated with increasing oxidative stress in hepatocytes. In normal circumstances, oxidation and antioxidation are kept in a dynamic balance. Oxidation can exceed antioxidation if it is influenced by infection disease or the environment.

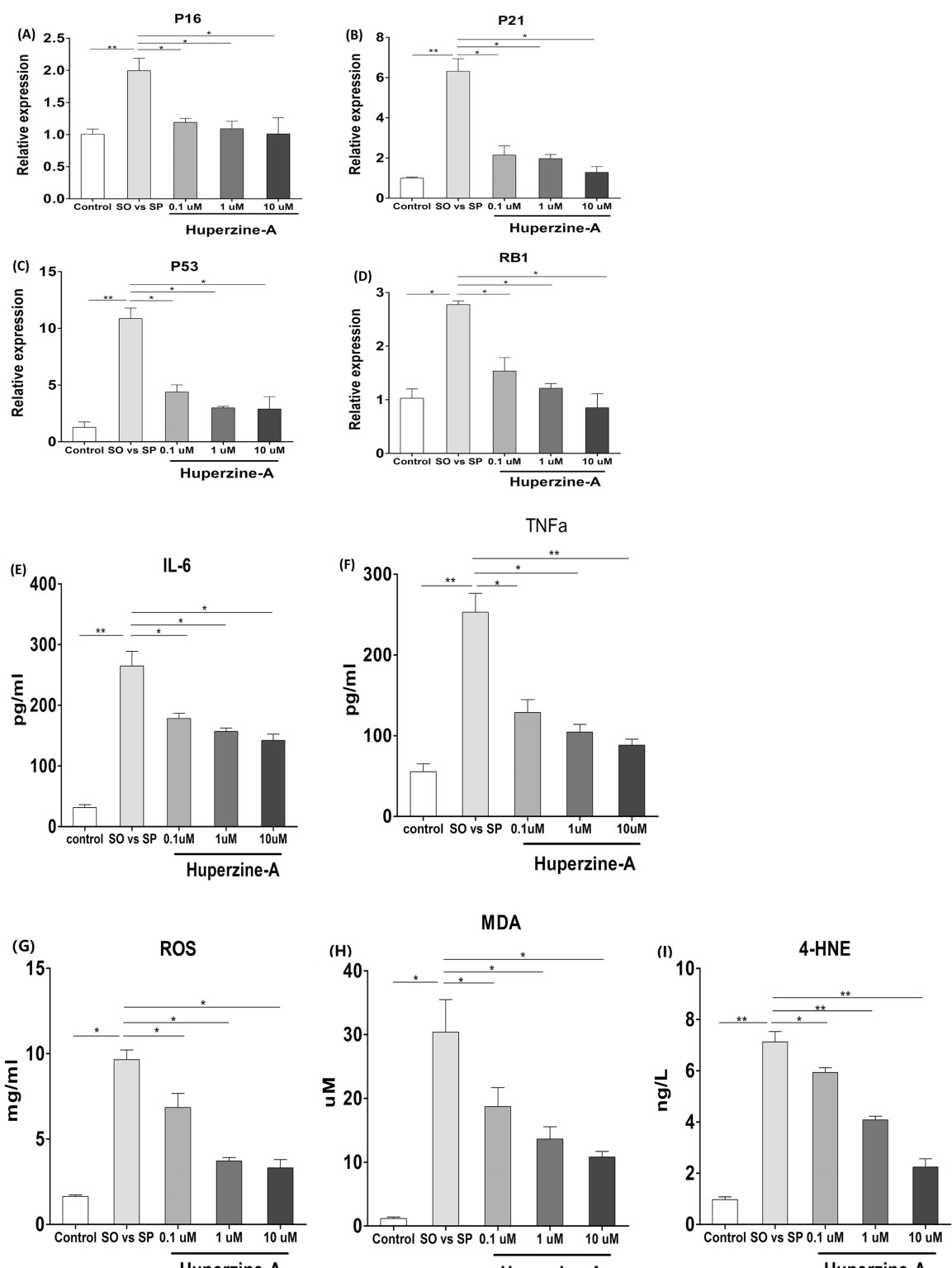
**Figure 4 The expression of senescence genes, inflamm-aging factors and oxidative stress.** (A–D) L-02 cells were treated with FFA, FFA + LH, FFA + MH and FFA + HH for 24 h and then the expression of p16, p21, p53 and pRb were determined by qRT-PCR. The expression of p16, p21, p53 and pRb in the FFA group was higher than in the control group ($P < 0.01$). All of the mRNA levels of above aging genes in the FFA + LH, FFA + MH and FFA + HH groups were decreased compared with those in the FFA group ($P < 0.01$). (E–F) The protein levels of TNF-$\alpha$ and IL-6 were detected by ELISA. The protein levels of TNF-$\alpha$ and IL-6 in the FFA group were higher than in the control group ($P < 0.01$). They were decreased in the FFA + LH, FFA + MH and FFA + HH groups compared with those in the FFA group ($P < 0.01$), but with no significant difference among FFA + LH, FFA + MH and FFA + HH groups ($P > 0.05$). (G–I) The expression of MDA, 4-HNE and ROS were determined by ELISA. The levels of MDA, 4-HNE and ROS were increased in the FFA group compared with those in the control group ($P < 0.01$), and they decreased in the FFA + LH, FFA + MH and FFA + HH groups compared with those in the FFA group ($P < 0.01$). In addition, they decreased with increasing huperzine A concentrations ($P < 0.05$). Data presented as the mean ± SE ($n = 3$ independent replicates). FFA mixture: 0.5 mmol/L, sodium oleate (SO)/sodium palmitate (SP) mixed at a 2:1 ratio; LH: FFA + 0.1 μmol/L huperzine A; MH: FFA + 1.0 μmol/L huperzine A; HH: FFA + 10 μmol/L huperzine A.

ROS accumulate and damage the organ, a process called oxidative stress. Oxidative stress induces the upregulation of inflammatory factors. This study found that the levels of oxidants, such as MDA, 4-HNE and ROS, were higher in the L-02 fatty liver cells. The inflammatory factors, such as TNF-$\alpha$ and IL-6, were increased in the L-02 fatty liver cells as well. The above results revealed that the in vitro NAFLD model also represents some aspects of hepatic steatosis.

Recent studies reveal that hepatocyte senescence predicts the progression of NAFLD (*Tachtatzis et al., 2015*). Hepatocyte senescence influences the fibrosis stage and an adverse clinical outcome (*Aravinthan et al., 2013*; *Tachtatzis et al., 2015*). Our study also documented that the proliferation activity of the L-02 fatty liver cells was inhibited. Levels SA-$\beta$-gal, a first and common marker of senescent cells, were higher in the L-02 fatty liver cells than in the normal cells. The expression levels of aging associated genes, such as p53, p21, p16 and pRb, were apparently elevated in the fatty liver cells. These results revealed that the fatty liver cells had a senescence phenomenon. However, how the hepatocytes become senescent in NAFLD is not very clear.

Studies report that oxidative stress plays an important role in cellular senescence and aging (*Zhang et al., 2017*; *Gambino et al., 2013*; *Venkatachalam, Surana & Clement, 2017*). Aging is delayed after inhibiting oxidative stress in mammals (*Gambino et al., 2013*). Oxidative stress causes double-strand DNA breaks, which, in turn, lead to permanent cell cycle arrest and cellular senescence (*Gambino et al., 2013*). Cellular senescence claims to lose the regenerative capacity of cells and recapitulate aging in vivo, which then leads to various diseases (*Campisi & D'Adda, 2007*). The activation of oxidative stress and inflammation leads to liver injury. Cell replication and regeneration are unique responses of the injured liver. However, some somatic cells cannot divide indefinitely and undergo replicative senescence (*Paradis et al., 2001*). Increased replicative senescence had been confirmed in chronic hepatitis C, cirrhosis and HCCs (*Brunt et al., 2007*). Recent studies have documented that there is a link between hepatocyte senescence and permanent cell cycle arrest in NAFLD (*Brunt et al., 2007*; *Aravinthan et al., 2013*; *Kim, Kisseleva & Brenner, 2015*; *Ping et al., 2017*). However, the relationship between oxidative stress and hepatocyte senescence in NAFLD is unclear.

Huperzine A is proved to have anti-inflammation and antioxidative effects in the immune and central nervous systems of mammals (*Tabet, 2006*; *Wang et al., 2008*).

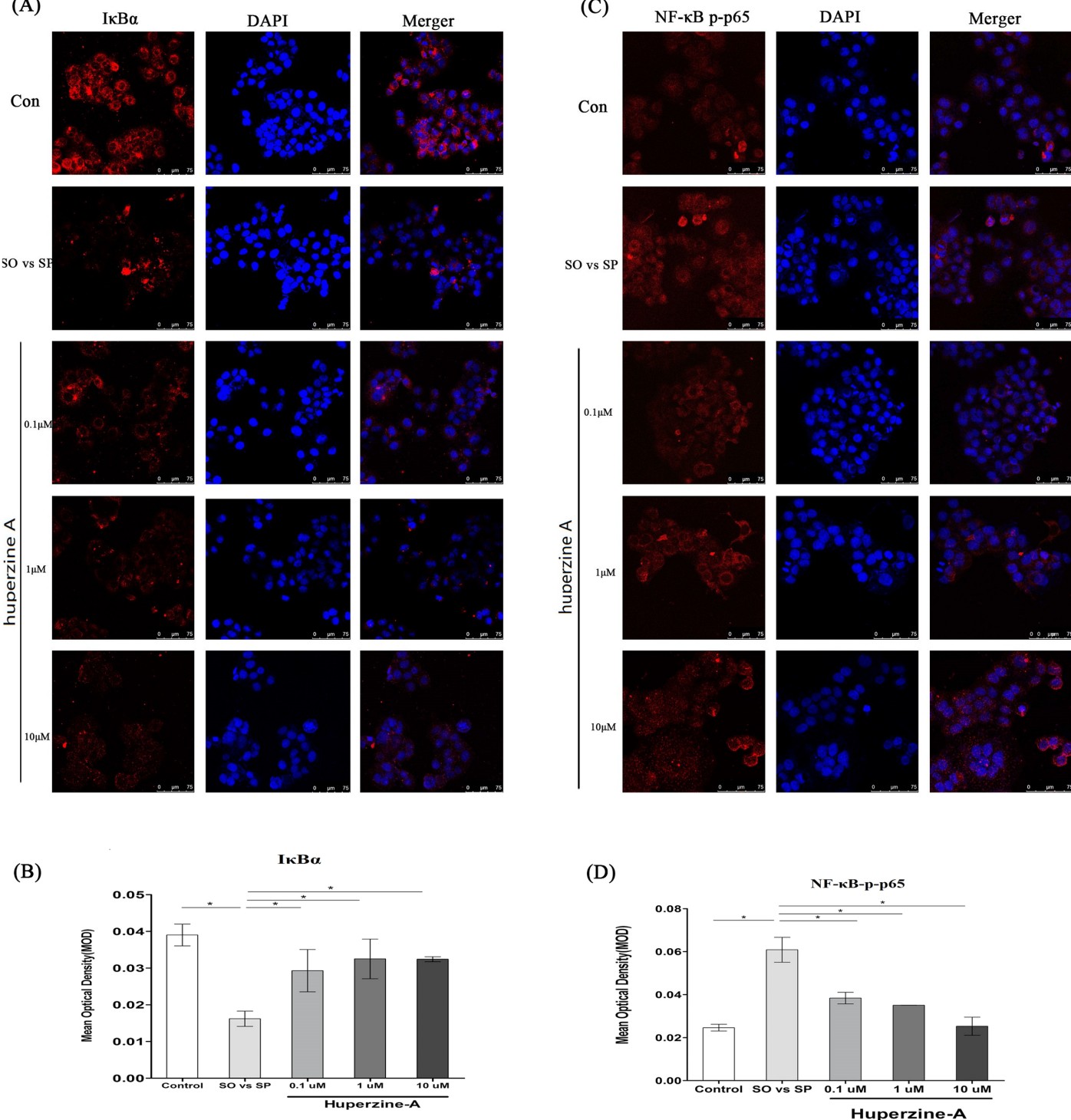

**Figure 5 The expression of NF-κB and IκBa.** (A) L-02 cells were treated with FFA, FFA + LH, FFA + MH and FFA + HH for 24 h. The expression of IκBa was determined by immunofluorescence assay. (B) The mean optical density (MOD) value of IκBα. It was decreased in the FFA group compared with that of the control group ($P < 0.01$), but increased in the FFA + LH, FFA + MH and FFA + HH groups compared with that in the FFA group ($P < 0.01$). (C) The expression of NF-κB p-p65 was determined by immunofluorescence assay. (D) The MOD value of NF-κB-p-p65. It was higher in the FFA group than in the control group, but was inversed in the FFA + LH, FFA + MH and FFA + HH groups than in theFFA group ($P < 0.01$). Data presented as the mean ± SE ($n = 3$ independent replicates). FFA mixture: 0.5 mmol/L, sodium oleate (SO)/sodium palmitate (SP) mixed at a 2:1 ratio; LH: FFA + 0.1 μmol/L huperzine A; MH: FFA + 1.0 μmol/L huperzine A; HH: FFA + 10 μmol/L huperzine A.        

Our previous study also showed that huperzine A attenuated oxidative stress, DNA damage and liver injury in D-galactose-treated rats (*Ruan et al., 2013*). In this study, we found that lipid droplets decreased after huperzine A treatment. Huperzine A decreased the levels of oxidants, such as MDA, 4-HNE and ROS, and inflammatory factors, such as TNF-α and IL-6. Huperzine A also decreased the expression levels of the genes p53, p21, p16 and pRb in L-02 fatty liver cells. All of the above evidence revealed that oxidative stress promoted hepatic cell senescence and that cell senescence was eased after the administration of huperzine A.

It has been reported that oxidative stress induces the expression of p53 (*Gambino et al., 2013*). The tumor-suppressor p53 activates a cellular response to DNA damage and results in a pause in cell proliferation by means of senescence or apoptosis (*Soussi & Beroud, 2001*). p53 induces cell cycle arrest via the p53/p21 pathway (*Fumagalli et al., 2014*). This transcriptional pathway involves the downregulation of a series of critical genes in G1/S transcription. The role of p21 in the cell cycle is mainly to inhibit CDK2, which phosphorylates the Rb protein. Phosphorylated Rb protein, an important regulator for G1/S transcription, combines with the transcriptional factor e2F and inhibits the DNA duplication of important genes. It is reported that p21 is upregulated in duplicate senescence. The cells with an excessive express of p21 enter cell cycle arrest (*Inoue et al., 2009*). Replicative senescence is inhibited by a deactivated p21 coding gene (*Brown, Wei & Sedivy, 1997*). A previous study demonstrated an increase in the number of p21-positive hepatocytes in middle-to-old-aged steatohepatitis and cirrhosis patients (*Aravinthan et al., 2013*). Studies report that p16 is upregulated in aging cells. Cell senescence can also be induced by the p16/pRb pathway. The p16 protein inhibits the phosphorylation of the Rb protein or promotes the degradation of pRb and then inhibits G1/S transcription (*Fahham et al., 2009*). Fat accumulation in hepatocytes induces oxidative stress. Long-term oxidative stress produces excessive ROS, which can destroy nuclear DNA and induce the DNA damage response (*Soussi & Beroud, 2001*; *Fumagalli et al., 2014*; *Inoue et al., 2009*) or directly regulate the aging-related signal pathway, thus promoting hepatocyte senescence.

In our study, we also found that the cell density was significantly decreased after FFA treatment compared with that of the control. The levels of apoptosis proteins, such as Bcl2, Bax, CytC and cleaved caspase 9, were higher in the fatty liver cells than in the control. As is well known, the acceleration of apoptosis is one feature of aging. It is reported that the expression levels of p53, p21 and Bax are increased in HepG2 cells during serum withdrawal-induced apoptosis (*Bai & Cederbaum, 2006*). p53 transfers to the mitochondria and then changes the mitochondrial membrane potential and induces the inter-reaction of the Bcl-2 family (*Kroemer, Galluzzi & Brenner, 2007*). Additionally, p53 upregulates the expression of pro-apoptotic proteins, such as Apaf-1, Bax, Fax, NOXA and PUMA, and downregulates that of Bcl-2. Other studies report that p21 is also an important factor in cell apoptosis. The expression of p21 is upregulated in osteoclast apoptosis induced by trichostatin A (*Yi et al., 2007*). The effect of Bcl-2 on anti-apoptosis is inferior to the effect of p21 on pro-apoptosis (*Marshall & Shankland, 2006*). There is a close association between p21 and Bax in cell apoptosis. p21 exhibits

its pro-apoptosis effect by changing the ratios of Bcl-2/Bax. p21 can also induce cell apoptosis via ROS accumulation (*Macip et al., 2002*). From our study, we found that FFA increased cell apoptosis. The expression levels of the apoptotic proteins, such Bcl2, Bax, CytC and cleaved caspase 9, decreased after huperzine A treatment. Huperzine A might promote cell apoptosis, alleviate cell senescence and inhibit the progression of NAFLD via its antioxidative effect.

Nuclear factor-κB is a nuclear transcription factor with a very wide function, regulating a series of genes, such as those of cytokines, inflammatory factors, immune related receptors and adhesion molecules (*Wang, Ma & Zhao, 2013*). Various endogenous and exogenous ligands, as well as mechanical and chemical stimulation, can activate NF-κB. The hyperactivation of NF-κB has been proven in numerous age-associated diseases, such as atherosclerosis, diabetes, neurodegeneration or immunosenescence (*Le Saux, Weyand & Goronzy, 2012*; *Tak & Firestein, 2001*). The NF-κB family of transcription factors are composed of five subunits, including RelA (p65), c-Rel, RelB, NF-κB1 (p105/p50) and NF-κB2 (p100/p52), in mammalian cells (*Ghosh et al., 2012*). The precursor proteins p100 and p105 function as IκBs, and they are transformed into short, active protein forms, such as p52 and p50, by a restrictive protease. Upon binding to (p65), c-Rel or RelB, the nuclear regulatory subunits p52 or p50 form heterodimers and activate downstream signal transduction pathways. In most normal cells, NF-κB is found in the cytoplasm of unstimulated cells in association with an inhibitory IκB protein. In the classical NF-κB pathway, the combination of ligands and cell surface receptors recruit downstream adaptor proteins (TRAF family protein or RIP), which recruit the IκB kinase complex and activate them, causing the degradation of IκBα, and this effect activates the transcription of genes downstream (*Sen & Smale, 2010*; *Hoesel & Schmid, 2013*; *Iannetti et al., 2014*). NF-κB transcription factors respond to pathogen attack as well as external and internal danger signals, including oxidative stress (*Perkins, 2007*; *Sfikas et al., 2012*; *Osorio et al., 2012*). The subunits of NF-κB2 and RelB in the alternative NF-κB pathway influence the expression of numerous key regulation genes of the cell cycle. NF-κB2 regulates the expression of CDK4 and CDK6, and RelB regulates the stability of p53 and p21WAF1 (*Perkins, 2007*). The above genes unite to regulate the activity of Rb and then lead to the expression of the polycomb group protein EZH2 (*Iannetti et al., 2014*; *Vaughan & Jat, 2011*). In our studies, we found that in the L-02 fatty liver cells, the expression of NF-κB was increased, while the expression of IκB was decreased. After huperzine A treatment, the expression of NF-κB decreased, while the expression of IκBα increased. NF-κB plays an important role in cell senescence in the fatty liver.

## CONCLUSION

In summary, the results of the present study proved that fatty liver cells might be liable to senescence. Oxidative stress may play an important role in promoting cell senescence in NAFLD. Oxidation resistance may promote cell apoptosis and inhibit cell senescence in NAFLD.

Huperzine A may inhibit the progression of NAFLD via NF-κB signaling. Further studies are needed to test hepatic cell senescence in animal models of NAFLD and in patients with NAFLD. Further studies about the critical genes involved in regulating hepatic cell senescence in NAFLD are also needed.

## ACKNOWLEDGEMENTS

We gratefully acknowledge Dr. Yan-guang Chen from Jiaotong University, who assisted with immunofluorescence and immunohistochemistry study. The authors are thankful to Dr. Xiao-feng Yu and Yi-qian Wang from Huadong Hospital, Fudan University, for their support in our endeavors.

### Funding

This work was supported by the National Natural Science Foundation of China (81701374), the Shanghai Municipal Commission of Health and Family Planning, Key developing disciplines (2015ZB0501), the Shanghai sailing program (No. 17YF1405200), the Shanghai Municipal Commission of Health and Family Planning research program (Youth Project, 20154Y0002), and the Shanghai Municipal Commission of Health and Family Planning research program (Surface project, 201440550). The funders had no role in study design, data collection and analysis, decision to publish, or preparation of the manuscript.

### Grant Disclosures

The following grant information was disclosed by the authors:
National Natural Science Foundation of China: 81701374.
Shanghai Municipal Commission of Health and Family Planning, Key developing disciplines: 2015ZB0501.
Shanghai sailing program: No. 17YF1405200.
Shanghai Municipal Commission of Health and Family Planning research program: Youth Project, 20154Y0002.
Shanghai Municipal Commission of Health and Family Planning research program: Surface project, 201440550.

### Competing Interests

The authors declare that they have no competing interests.

### Author Contributions

- Xiao-na Hu conceived and designed the experiments, performed the experiments, contributed reagents/materials/analysis tools, approved the final draft, submission, revised.
- Jiao-feng Wang performed the experiments, analyzed the data, contributed reagents/materials/analysis tools.
- Yi-qin Huang performed the experiments.

- Zheng Wang prepared figures and/or tables, revised.
- Fang-yuan Dong prepared figures and/or tables.
- Hai-fen Ma prepared figures and/or tables.
- Zhi-jun Bao conceived and designed the experiments, authored or reviewed drafts of the paper.

## Data Availability

The raw data are provided in the Supplemental Files.

## Supplemental Information

Supplemental information for this article can be found online at http://dx.doi.org/10.7717/peerj.5145#supplemental-information.

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
