# Peer review of "Huperzine A attenuates nonalcoholic fatty liver disease by regulating hepatocyte senescence and apoptosis: an in vitro study"

_PeerJ, doi:10.7717/peerj.5145_

## Round 0.1 · original submission · Major Revisions

I agree with the reviewers that there are several areas where authors must improve description of methods, experimental design, etc. I also suggest that authors seek professional editorial help to resolve issues with grammar and sentence structure.

·

Basic reporting

no comment

Experimental design

no comment

Validity of the findings

no comment

Additional comments

This manuscript addressed an important issue of huperzine A on oxidative stress and hepatocyte senescence in nonalcoholic fatty liver disease. Xiao-na Hu et al. used an in vitro fatty liver model. Data were collected on inflammatory factors, oxidation product and hepatocyte senescence. However, the manuscript cannot be accepted in its current form because it needs to be improved.

1. There are repetitions in the manuscript that needs to be addressed.
2. Figure1. The cell viability was lower in L02 cells treated with different ratio and concentrations fatty acid compared with control. What explains this? Is the concentration of fatty acids too high? Need to clarify.
3. How do you treat with fatty acids? It needs to be described in the Materials and Methods.
4. Which fatty acid concentration is used in Figure 2, 3, 4. It needs to be described in the Figure Legends.
5. Figure 2A needs to add scale.
6. Figure 2D. It's better to show the protein expression pattern.
7. GAPDH was as reference gene in your experiment, whether is suitable or not for hepatocytes. Do you have basis of some scientific researches?

Reviewer 2 ·

Basic reporting

The manuscript was not written in professional English. There are various grammatical and non-scientific terms used throughout the manuscript. Authors should consult professional academic writing service.

The authors did not show sufficient background in the field.

Figures are in poor quality and really confusing. Be more informative in legends and state clearly what the units are for each measurements.

Experimental design

Methods are extremely ill defined. It is questionable how authors obtained their results.

Research question is not meaningful due to poor experimental design. Authors should consider re-run the experiment with better experimental design.

Validity of the findings

Data generated are not trustworthy due to problematic methods.

Additional comments

The study is poorly designed and the manuscript is not well written. There are major flaws in this study in addition to academic writting and therefore should not be considered for publication.
Major issues:
1. The author used fatty acid without conjugating to BSA, which do not represent in vivo scenario.
2. Authors used a certain ratio of fatty acids without justification of the composition and ratio or its physiological relevance. Yet the authors claimed their model was successful just because there is more lipid accumulation in the cell compared with Control.
3. Also, the author used a fetal hepatic cell line to study adult fatty liver disease.
4. The methods are poorly defined.
5. Extremely casual results interpretation without scientific rigor and thus not trustworthy. For instance, authors claimed cell senescence are altered in response to Huperzine A treatment, from Fig2, it was clear that the cell density are different between treatments. The authors disregard such pitfalls and just interpret the result to their favor.

---

## Round 0.2 · Major Revisions

There is merit in the work reported here, but authors need to make an extra effort in improving writing style. In addition, please address the specific points raised by the reviewer within the appropriate section in the manuscript. Unless these issues are taken into account during revision, it will not be possible to consider the manuscript further.

Reviewer 2 ·

Basic reporting

English has been improved grammatically, but still not academic writing level.

Experimental design

See specific comments

Validity of the findings

See specific comments

Additional comments

General comments:
The authors need to write up to scientific standard. When reporting results, every time the author claims significance or difference, statistical proof is required. Also, Materials and methods were mixed in the results section. The authors need to restrain from making general statements. Only suggest and conclude based on what your results tell you, not generalize.

Ln65-66: duplicated reference
Ln67: studies? Only 1 reference is shown.
Ln83: What is a super net?
Ln170: If authors chose to use 2- Ct method, PCR efficiency needs to be included.
Ln200-203: Describe the model used. Including fixed and random effects.
Ln209: Where’s statistical proof of difference?
Ln210: where’s your statistical proof of your claimed significance?
Ln216-218: This is not results. Move to Materials and methods section.
Ln225: avoid using words such as “obvious” in here throughout the manuscript.
Ln290-294: repetition of Ln 220-224.
Ln301: The author’s model seemed to be able to imitate some aspects of the steatosis, but authors cannot claim that the model is well established. Two-hit model is already outdated, authors should acknowledge the fact that multiple-hit model better reflect NAFLD. Therefore, authors should instead state here that the model in this study represents some aspects (lipid accumulation, inflammation) of hepatic steatosis.
Ln 304-305: This study or further studies? If further studies, where are your references?
Ln310: Which studies? Where are your references?
Ln 320: This is your objective of this study? If so, relocate it to the end of the introduction section.
Ln322: new? Specify what new anti-inflammation and antioxidative effects?
Ln323: studies? Why is there only 1 reference?
Ln329: The authors need to restrain from making general statements. Only suggest and conclude based on what your results tell you, not generalize. You only tried Huperzine A here, you do NOT have evidence to suggest other antioxidative agents’ effect.

---

## Round 0.3 · Minor Revisions

Please include the actual values for amplification efficiency (page 6). It is unclear what is meant by "confirmed to be consistent...". The actual values must be reported.

Page 7, data "are" not "were". Also separate "groups" from "were". They're running together in the current version
Page 9, "suitable" instead of "well" when referring to the NAFLD model
Page 9, "The above results......in vitro..." not "vitro"

---

## Round 0.4 · accepted · Accept

Authors have addressed all comments from reviewers. The paper is now acceptable.

#